# Therapies Based on Adipose-Derived Stem Cells for Lower Urinary Tract Dysfunction: A Narrative Review

**DOI:** 10.3390/pharmaceutics14102229

**Published:** 2022-10-19

**Authors:** Meng Liu, Jiasheng Chen, Nailong Cao, Weixin Zhao, Guo Gao, Ying Wang, Qiang Fu

**Affiliations:** 1Department of Urology, Shanghai Sixth People’s Hospital Affiliated to Shanghai Jiao Tong University School of Medicine, Shanghai Eastern Institute of Urologic Reconstruction, Shanghai Jiao Tong University, Shanghai 200233, China; 2Wake Forest Institute for Regenerative Medicine, Winston-Salem, NC 27157, USA; 3Key Laboratory for Thin Film and Micro Fabrication of the Ministry of Education, School of Sensing Science and Engineering, School of Electronic Information and Electrical Engineering, Shanghai Jiao Tong University, Shanghai 200240, China

**Keywords:** adipose-derived stem cells, stem cells therapy, lower urinary tract dysfunction

## Abstract

Lower urinary tract dysfunction often requires tissue repair or replacement to restore physiological functions. Current clinical treatments involving autologous tissues or synthetic materials inevitably bring in situ complications and immune rejection. Advances in therapies using stem cells offer new insights into treating lower urinary tract dysfunction. One of the most frequently used stem cell sources is adipose tissue because of its easy access, abundant source, low risk of severe complications, and lack of ethical issues. The regenerative capabilities of adipose-derived stem cells (ASCs) in vivo are primarily orchestrated by their paracrine activities, strong regenerative potential, multi-differentiation potential, and cell–matrix interactions. Moreover, biomaterial scaffolds conjugated with ASCs result in an extremely effective tissue engineering modality for replacing or repairing diseased or damaged tissues. Thus, ASC-based therapy holds promise as having a tremendous impact on reconstructive urology of the lower urinary tract.

## 1. Rational for Focusing on ASCs

Lower urinary tract dysfunction, which can be caused by various factors such as aging and trauma, is not usually directly life-threatening. However, it often greatly reduces a patient’s quality of life and brings a heavy economic burden to individuals and society.

Traditional treatment methods often use autologous tissue or synthetic materials for repair and reconstruction, which may lead to in situ complications and immune rejection. Stem cell therapy that can repair damaged tissues is attracting increasingly more attention. The core concept of stem cell therapy is to apply cell self-replication, cell differentiation, and paracrine mechanisms to facilitate regeneration and restore the function of damaged host tissues [1,2]. Stem cells are mainly divided into embryonic and post-natal stem cells. Because of the ethical obstacles and uncertain carcinogenicity of embryonic stem cells, the use of these cells is currently limited. Post-natal stem cells are relatively safe and controllable. Stem cells can be obtained from a variety of different tissues and organs, such as skeletal muscle, bone marrow, adipose tissue, and blood [3,4], Bone marrow-derived stem cells, as the first batch of recognized stem cells, have been used in the urinary system for regenerative therapy for many years, but because the bone marrow-derived stem cells need to be extracted from autologous bone marrow, it is a traumatic and painful process for patients. The incidence of in situ complications is also relatively high, while muscle-derived stem cells also have the same problem [5]. Although urine-derived stem cells are cost-effective, there have been relatively few studies in this area [6].

Adipose-derived stem cells (ASCs) have great potential because sufficient cell numbers can be obtained through liposuction, with less trauma to the patient and no serious in situ complications (Figure 1). The immunogenicity of ASCs is also relatively low and less likely to cause immune rejection [7]. ASC-based regenerative therapy mainly relies on two different approaches to restore the structure and function of lower urinary tract tissues: (1) Injection of autologous or allogeneic ASCs or their secreted exosomes to promote tissue regeneration. Exosomes are vesicles, with a diameter of about 40–100 nm, with a lipid bilayer membrane that can be released by many cells (In the early stage, the cell membrane is invaginated to form the initial endosome, and then bioactive substances start to accumulate in the early sorted endosome. Early endosomes then become late sorting endosomes under the control of the endocytic sorting complex and other related proteins required for trafficking. They eventually form multivesicular bodies after the second indentation. After the multivesicular body fuses with the cell membrane, the substances inside the cells are released to the outside in the form of vesicles, which are exosomes. The International Society for Extracellular Vesicles recommends that two types of proteins (e.g., transmembrane or glycolphosphatidylinositol-anchored proteins associated with the plasma membrane and/or endoderm, cytoplasmic proteins recovered in extracellular vesicles) need to be assessed to determine whether they are exosomes.) [8]. These vesicles can facilitate processes through mediating communication between cells. Exosome-depleted FBS MSCM (10%) is applied to fourth-generation ASC culture for 3 days, and then the supernatants are collected to extract the exosomes through ultracentrifugation; and (2) ASC-based tissue engineering constructs for reconstructive therapy; that is, implantation of natural or artificial biomaterials, combined with induced or uninduced ASCs or their exosomes, to promote and guide tissue repair. The application of ASCs is considered a promising therapeutic strategy for lower urinary tract repair and reconstruction. It is expected to be able to truly repair and restore the function of damaged tissue This article will review the repair and reconstruction of the lower urinary tract using ASCs.

## 2. Advances in ASCs Therapy for the Lower Urinary Tract

### 2.1. ASC-Based Therapy for Bladder Reconstruction

Repair and reconstruction of the bladder are still major challenges in the urology field. Gastrointestinal segment replacement therapy often causes a series of complications, including recurrent urinary tract infections, electrolyte imbalances, mucus production, and even malignancy, due to its incompatibility with urine and the characteristics of the gastrointestinal segment itself [9,10,11]. Recently, research on different types of stem cells for bladder reconstruction has become more common. Among these stem cell types, ASCs are considered an ideal cell source because of their easy availability with fewer complications. Their efficacy in bladder regeneration has also been proven in animal models, and increasingly more studies are now aimed at optimizing their therapeutic effects [12].

The cellular scaffold strategy involves in vitro expansion of autologous patient cells and seeding into bioengineering materials, followed by complete regeneration of bladder tissue in vivo [13]. Over the past few years, increasingly more studies have attempted to use acellular matrix grafts, which have the advantages of good biocompatibility, low immunogenicity, and excellent mechanical properties, and were shown to fully restore bladder capacity in 30–40% of bladder defects [14]. Zhe et al. [15] investigated the efficacy and mechanism of CM-DiI-labeled autologous ASC-seeded bladder acellular matrix grafts (BAMGs) for bladder repair. In this process the tissue is trimmed, and 0.25% trypsin is used to completely remove adipose tissue and fascia from bladder samples; subsequently, bladder samples are rinsed with PBS for 2 h and soaked in 0.2% Triton X-100 and 0.1% (v/v) ammonium hydroxide for 12 days. ASC-seeded BAMGs were cultured intraperitoneally in rats for 2 weeks before bladder repair. After 2 weeks, histological analysis showed significantly increased collagen bundle growth; labeled myofibroblasts and mesothelial cells were found both inside and on the surface of the scaffold, indicating that ASCs were efficiently differentiated into myofibroblasts and mesothelial cells. At 4 and 14 weeks after implantation, maximum bladder capacity was measured after surgery and prior to killing via an intravenous catheter inserted into the urethra and was defined as the volume that would leak out of the external urethral orifice after emptying the bladder. The results showed that the bladder capacity of the treatment group was significantly greater than that of the control group, and the results of immunofluorescence confirmed that the regeneration of smooth muscle cells was significantly accelerated after ASC-seeded BAMG treatment. The treatment group also showed more nerve cell regeneration 14 weeks after implantation. It may be more convincing if this study had added a comprehensive cystometric analysis. The experimental group showed an increase in bladder capacity at both 4 and 14 weeks postoperatively; however, the experiment did not provide detailed information such as bladder compliance and detrusor pressure to assess bladder function. Urodynamic analysis should be performed in further studies to accurately compare the therapeutic efficiency. To make up for the fact that most of these studies were performed in small rodents or rabbits, Pokrywczynska et al. [16] and Hou et al. [17] used similar methods in pigs and canine models, respectively, and obtained comparable results. Nevertheless, large animal models may encounter problems with incomplete regeneration of the central graft region, which may be related to insufficient graft revascularization. A study by Moreno-Manzano et al. [18] further illustrated this problem. They also used ASC-seeded BAMGs for bladder reconstruction. Histological results suggested that 10 days after surgery, smooth muscle actin at the implantation site was highly expressed in longitudinal and transversal layers, p63- and cytokeratin-7-positive urothelium was restored, and vimentin mesoderm marker expression was significantly increased in the submucosal mucosa. Immunohistochemical results showed that ASC-seeded BAMGs also accelerated angiogenesis and neuronal regeneration according to the distribution of CD31 and S100b. Together, these results suggest that ASCs play a crucial role in bladder regeneration.

In addition to ASC-seeded BAMGs, silk fibroin is widely used in tissue repair and reconstruction because of its good biocompatibility, excellent mechanical strength, and controllable biodegradation rate. Nevertheless, one study demonstrated that simple silk fibroin structures may increase the risk of urolithiasis and urinary leakage [19]. Xiao et al. [14] constructed a scaffold with better mechanical properties by prefabricating bilayer BAMGs and silk fibroin and then seeding ASCs to treat a rat model of bladder defects. The cytocompatibility of the material was excellent, and the mass seeding of ASCs did not significantly change the morphological or mechanical properties of the material. Implantation did not cause serious adverse reactions or metabolic disturbances. Histological examination revealed that after 2 weeks, smooth muscle fibers began to gradually replace collagen fibers and develop into muscle bundles with clear structures. Nerve cell numbers and blood vessel numbers and diameters significantly increased. ASCs promoted angiogenesis and innervation in regenerating bladder tissue, and urodynamic testing also indicated the significant recovery of bladder function. Furthermore, the expression levels of SDF-1α, VEGF, and their receptors were significantly upregulated, and ERK 1/2 phosphorylation was increased. These mechanistic pathways may accelerate angiogenesis. Experiments that balance the interrelationship between VEGF-mediated angiogenesis and AKT activation may complement the results. Whether these two pathways are synergistic or independent of each other remains to be addressed. The maturity of neovascularization and the degree of reinnervation have always been the main issues to be overcome for the clinical use of ASC-seeded BAMGs. In addition, the ladder wall tension, uniaxial tension test, and video cystogram performed in larger animals and larger samples may be good complements.

As a cross-linked polymer network, hydrogel is suitable for encapsulating cells because of its ability to absorb a large amount of water, its low toxicity, and its high cytocompatibility. Currently, it is one of the most popular biomaterials for encapsulating and delivering stem cells to target tissues [20]. In addition to silk fibroin, BAMGs that are supported by alginate hydrogel show good biocompatibility and excellent mechanical properties. Xiao et al. [21] used an alginate di-aldehyde-gelatin hydrogel-silk mesh combined with ASC-seeded BAMGs to build a tri-layer “sandwich” scaffold. Histological and functional assessments were performed at 2, 4, and 12 weeks after surgery. Smooth muscle regeneration, angiogenesis, and innervation of the bladder were significantly accelerated, which together promoted the recovery of bladder function. Interestingly, ASCs can also accelerate scaffold degradation to a certain extent and reduce the occurrence of fibrosis and inflammation. Xiao et al. [22] made further attempts and wrapped the ASC-seeded scaffold in omentum to promote neovascularization before bladder repair. ASCs exerted pro-angiogenic effects through the SDF-1α/CXCR4 pathway and its downstream ERK1/2 phosphorylation. In these two experiments, inadequate neuronal regeneration limited the diastolic as well as contractile function of the regenerated smooth muscle, which affected the complete recovery of bladder physiology after repair. In terms of materials, silk fibroin provides good mechanical properties and reduces the difficulty of surgery, but its own degradation also brings the risk of stones. On the one hand, together with the omentum, it can promote scaffold vascularization, and on the other hand, elucidating the mechanism may also make the experiment more complete.

Retention of the extracellular matrix (ECM) may increase the adhesion and expansion of ASCs. To address this, Wang et al. [23] designed and used a cell sheet treatment strategy to preserve the ECM and intercellular interactions and added bioengineered materials to enhance mechanical strength. Wang et al. [23] developed a bioengineered cell sheet for bladder repair and reconstruction, by building a scaffold using porous polyglycolic acid and adding an ultra-small superparamagnetic iron oxide nanoparticle-labeled ASC sheet to form a three-dimensional bladder patch. The ASC sheet transferred to the porous scaffold may act as a waterproof barrier, preventing urine penetration and indirectly reducing the inflammatory response. An intact ASC sheet can act as a flat, smooth surface for the bladder patch, thereby hindering urine crystallization or precipitation. Using immunofluorescence, they showed that the treatment significantly promoted urothelial cell, smooth muscle cell, and neurovascular regeneration. Urodynamics also showed increased bladder capacity and significant functional improvements. Although the combination of ASC sheets and porous scaffolds is a promising regenerative treatment for bladder injury, it cannot be ignored that the material itself has limitations. Simple porous scaffolds have the possibility of increasing the risk of mortality in rats by directing penetration of urine into the abdominal cavity; thus, future improvements in the materials and confirmation of the efficacy of bioengineered bladder patches in clinical applications are also required.

Yang et al. [24] evaluated the effects of ASC–exosome-stimulated BAMGs on promoting bladder tissue regeneration and functional recovery in a rat model. Compared with simple BAMG transplantation, angiogenesis was more pronounced with ASC–exosome-stimulated BAMGs, and ELISA analysis suggested that the significantly upregulated expression levels of VEGF and PDGF may be the underlying mechanism. Furthermore, NFκB inhibitors can reverse this process, indicating that the activation of this pathway depends on increased NFκB expression. Urodynamic tests revealed that bladder compliance, mean intravesical pressure, and mean bladder capacity were significantly increased in rats treated with ASC–exosome-stimulated BAMGs, indicating bladder function recovery. Immunofluorescence revealed that the combined treatment of ASC–exosome-stimulated BAMGs exhibited a higher density of CD31- and α-SMA-positive cells and a lower density of collagen fibers compared with BAMGs alone. To further clarify the mechanism of ASC exosomes in bladder remodeling, Xiao et al. [25] optimized BAMGs to form a three-layer hydrogel scaffold using alginate dialdehyde-gelatin hydrogel to encapsulate ASC–exosomes. After treatment with ASC–exosomes, the regeneration of blood vessels and smooth muscle was significantly accelerated, and inflammation and fibrosis were reduced. Although no significant change in bladder compliance was observed, urodynamic testing indicated an increased volume and restored function. Immunofluorescence showed that ASC–exosomes could enhance the proliferation and tube-like structure formation of umbilical vein endothelial cells by activating the CXCR4/SDF-1α pathway. By upregulating *miRNA-12* expression levels, inhibiting RGS16 to activate the CXCR4/SDF-1α pathway ultimately increased VEGF secretion. This study well-evaluated the feasibility of a three-layer composite scaffold composed of ASC–exosomes, BAMGs, and alginate dialdehyde-gelatin hydrogel for bladder repair by promoting angiogenesis. However, the angiogenic potential of other stem cells is not excluded in the repair process. Furthermore, it can also be beneficial to the neuroregeneration of the bladder by manipulating the ASCs to enhance the neurotropic potential of exosomes. In addition, there is still evidence that ASC–exosomes isolated from obese or diabetic donors are suboptimal in promoting angiogenesis. Therefore, experiments may need to further clarify the inclusion criteria of donors to reduce differences in donor-related effects. Table 1 is a summary of ASC-based therapy for bladder reconstruction.

### 2.2. ASC-Based Therapy for Urethral Injuries

Although modern urology has made great progress, the treatment of complex and long urethral injuries and strictures remains a huge challenge. Current surgical treatment methods are faced with difficult surgery, tissue fibrosis, chronic inflammation, poor angiogenesis, and other problems that can eventually lead to secondary stenosis [26,27]. At present, the amount of research on stem cell transplantation for treating urethral injury and delaying fibrosis has been increasing. ASCs have emerged in increasingly more experiments because of their easy availability, their potential for multi-directional differentiation, and their ability to regulate wound healing through paracrine effects [28].

Castiglione et al. [29] assessed whether local injection of ASCs could prevent urethral fibrosis in a rat model. Fibrosis-inducing transforming growth factor (TGF)-β1 was injected into the urethral wall after incision. After 1 day, ASCs were injected into the urethral wall for treatment. After 4 weeks, rats were subjected to histological and functional assessments. Urodynamic tests suggest that the voiding interval was prolonged by 49% in the treatment group, and the single voiding volume, urine flow rate, bladder compliance, and bladder volume were significantly increased. In the treatment group, no urethral wall thickening was observed, and the lumen was unobstructed. Western blotting results indicated that collagen III, collagen I, and elastin were closer to normal levels after treatment than in the control group. Furthermore, five fibrosis-associated genes were significantly reduced. The application of xenogeneic stem cells in the treatment of early urethral strictures has achieved good results. From the perspective of safety, autologous stem cells may be a more mainstream research direction at present. Importantly, most patients with urethral strictures and fibrosis are already in the advanced and chronic stages of the disease when they visit the clinic. If the experiment can further evaluate the therapeutic effect of ASCs on established and recurrent urethral fibrosis, it may be more clinically meaningful.

Several studies have shown that *miR-21* can promote the proliferation, differentiation, and paracrine effects of mesenchymal stem cells, and thus it has great potential in regenerative medicine. *MiR-21* has also been recently shown to have a critical role in the regulation of skin fibrosis [30,31,32]. To date, studies have attempted to combine *miR-21* and ASCs for the treatment of urethral fibrosis. Feng et al. [28] explored the possibility that *miR-21* could enhance the effect of ASCs in preventing urethral fibrosis. *MiR-21*-modified ASCs were constructed by lentivirus-mediated transfer of pre-*miR-21* and *GFP* reporter gene and injected locally into the urethral wall. In vitro, *miR-21* modification had no obvious negative effect on ASC viability but significantly increased the expression levels of the angiogenic factors hypoxia inducible factor-1, vascular endothelial growth factor (VEGF), basic fibroblast growth factor (bFGF), stem cell factor, and stromal derived factor-1a. In animal experiments, *miR-21*-modified ASCs could scavenge oxygen free radicals in the wound microenvironment, improve the survival rate of stem cells, and accelerate the formation of new blood vessels by regulating the expression levels of the abovementioned six angiogenic factors. Overall, this modification improved the therapeutic potential of ASCs to prevent fibrosis. Modification of ASCs with *miR-21* can increase the therapeutic potential of ASCs against urethral stricture formation, but its long-term therapeutic effect requires further validation. Furthermore, the main purpose of this study was to test the feasibility of *miR-21* modification to enhance the therapeutic potential of MSCs for postoperative urethral healing and reconstruction, which may be used to limit the recurrence of urethral strictures after urethrotomy. However, it remains to be tested if local injection of ASCs, especially genetically modified ASCs, can be used to treat urethral strictures that have already formed.

Constructing urethral tissue using self-assembly scaffolds is also a promising method that could finally solve the problem of urethral injury. Auger et al. [33,34] first introduced the construction of self-assembly scaffolds to replace human connective tissue in 1997, and Magnan et al. [35] fully demonstrated its possibility in the urogenital field in 2009. Compared with traditional dermal fibroblasts, ASCs have better regenerative potential and do not promote fibrosis or lead to scarring. From this point of view, the ASC-assembly scaffold is more advantageous in repairing urethral tissue [36]. Rashidbenam et al. [37] co-cultured ASCs with vitamin C to stimulate ECM production. Then, the ASC sheet seeded with urothelial and smooth muscle cells was used as a scaffold. After 14 days in culture, the urothelial cells on the cell sheet continued to proliferate and express CK7, CK20, UPIa, and UPII, and smooth muscle cells also proliferated normally and maintained the expression of smooth muscle actin (SMA), myosin heavy chain, and smoothelin. This study clarified the possibility of artificially constructing and stacking a urethra with a complete structure; however, relevant animal experiments are lacking. Zhou et al. [38] further refined this idea and demonstrated for the first time that a complete three-layer tissue-engineered urethra can be successfully constructed using cell sheet technology and can promote the restoration of urethral structure and function. Using a similar technique, a three-layer tissue-engineered urethra was built and labeled with ultra-small superparamagnetic iron oxide and implanted subcutaneously for 3 weeks to facilitate graft vascularization. Even 3 months after urethral replacement, the tissue-engineered urethra labeled with ultra-small superparamagnetic iron oxide could still be effectively detected by MRI. Furthermore, the three-layer structure of the bioengineered urethra was intact, and the blood vessel density was significantly increased and approached normal levels. The tissue-engineered urethra using cell sheet technology still faces the problems of a long development cycle and complicated procedures. In addition, the injury model established by the experiment may have more clinical significance if long-segment urethral injuries larger than 2 cm are included for further evaluation.

Preventing fibrosis and reducing scarring have a crucial role in urethral repair. Several studies have suggested that TIMP-1, which is highly expressed in urethral scar tissue, plays a crucial role. Sa et al. [39] pre-epithelialized ASCs in vitro, whereby ASCs were seeded on the upper side of the membrane of a Millicell insert coated with 0.10% collagen type IV. The culture medium was DMEM supplied with 2% FBS, 20 ng/mL epidermal growth factor, 2.5 mM all-trans retinoic acid, 10 ng/mL keratinocyte growth factor, 0.5 mg/mL hydrocortisone, and 10 ng/mL hepatocyte growth factor. When they were seeded with BAMGs, fibrosis in urethral tissue was successfully reduced, resulting in a wider urethra with less scar tissue. Further research found that post-transcriptional inhibition of TIMP-1 by *miR-365* in ASCs could further inhibit fibrosis and improve the effect of ASC transplantation in the treatment of urethral injury. The failure of urethral reconstruction is often associated with the limited regeneration of epithelial tissue. Previous studies have confirmed that the use of in vitro culture-expanded epithelial cell composite scaffolds for urethral repair can promote urothelial proliferation in the repaired segment. However, epithelial cells obtained from autologous epithelial tissue have limited material sources, and in vitro culturing and expansion are prone to aging, making it difficult to achieve the numbers of cells required for the composition of scaffold materials. Li et al. [40] pre-epithelialized ASCs and then seeded them onto BAMGs to treat urethral defects, achieving better therapeutic effects. Histological findings indicated that urethral continuity was restored, and the formation of a continuous epithelial layer with a localized multilayered structure was observed in the early stages after implantation. This suggests that pre-epithelialized ASCs can be a potential cell source for urothelial tissue engineering. Although the rabbit ASCs used in this experiment have a high degree of homology with human ASCs, the main process of the experiment is to use ASCs for treatment after epithelial induction in vitro. Whether the difference between the primitive phenotypes of the two cells could be an influencing factor in cell differentiation is currently inconclusive; optimizing the complex design of in vitro induction to achieve mature epithelial differentiation of ASCs is also the direction. Fu et al. [41] further combined the myogenic differentiated ASCs induced by mechanical extension and oral mucosal epithelium to form an artificial urethral substitute with a complete double-layer structure. They used poly-glycolic acid as a scaffold for the treatment of urethral defects and achieved better therapeutic results. In this experiment, it was found that mechanical extension had a positive effect on the construction of tissue-engineered urethra, but only one extension strength was used, and an optimal mechanical stimulation mode could not be accurately obtained. The artificial urethra substitute constructed in the experiment was divided into two layers. If it could improve the relatively low density of smooth muscle cells in the muscle layer, it may achieve better therapeutic effects.

Silk fibroin hardly causes any immune responses after purification and silk sericin removal. It can promote cell adhesion and proliferation, does not cause chronic inflammation (which exacerbates the risk of fibrosis), and is being investigated as a suitable bio-scaffold [42,43,44]. Tian et al. [45] used BrdU-labeled ASCs and seeded silk fibroin to repair urethral defects. At 6 weeks after surgery, 6–7 layers of intact and regularly arranged urothelial cells were formed on the surface of the silk fibroin material, and the numbers of smooth muscle cells and new blood vessels growing along the silk fibroin pores were significantly increased. The expression levels of factor-VIII-related antigen and α-SMA were significantly increased. Pan-cytokeratin staining was positive, and the cytoplasm was uniformly brown and reticular under high magnification, which is more like normal urethral mucosa. Interestingly, the degradation rate of silk fibroin material was also accelerated. 

The use of exosomes for cell-free therapy of urethral injuries is also increasing. Wang et al. [46] evaluated the role of ASC exosomes encapsuled by a collagen/poly(L-lactide-cocaprolactone) nanoyarn scaffold in promoting the repair of urethral injury and inhibiting fibrosis. The ASC exosome nanoyarn scaffold significantly promoted epithelialization and vascularization and accelerated the transition of damaged tissues from an inflammatory state to a regenerative state. This treatment does not promote fibroblast hyperproliferation or collagen I expression, and thus represents an effective cell-free treatment strategy. Table 2 is a summary of ASC-based therapy for urethral injuries.

### 2.3. ASC-Based Therapy for Stress Urinary Incontinence (SUI)

SUI affects the quality of life of patients and has brought a huge economic burden to health systems around the world. Both sling and artificial sphincters are effective in treating SUI by correcting the poor anatomical support of the urethra. However, the fundamental factor leading to SUI, namely the basic problem of functional decline caused by atrophy of the urethral sphincter, has not been well resolved and the long-term postoperative effect is unsatisfactory [47,48,49]. In the past decade, regenerative methods based on stem cell injection have attracted increasing attention. Such methods can promote the proliferation of urethral sphincter muscle cells and essentially restore the structural and functional defects of the urethral sphincter, representing ideal alternative therapeutic strategies for treating SUI [9].

Previous studies have shown that co-culturing stem cells can enhance their potential to proliferate and differentiate [50,51]. Tehrani et al. [52] co-cultured autologous ASCs and muscle-derived stem cells and then implanted them into a SUI rat model through intraurethral injection. This obtained better therapeutic effects than a single type of stem cell treatment. The SUI model was constructed using bilateral pudendal nerve dissection. Three weeks after co-injection of ASCs and muscle-derived stem cells, urodynamics, histology, and immunohistochemistry were performed. Compared with the single stem cell treatment group, the striated muscle and external sphincter around the urethra were significantly thickened, and the urethral pressure curve was significantly higher in the combined treatment group. The researchers suggested that the improved treatment efficacy may be attributable to a more comprehensive paracrine factor and cytokine milieu brought by the combined therapy. The experiments detailed stem cell isolation and culture techniques, but no immunohistochemical assessment was performed to differentiate the cultured cell populations, which may be a point of improvement.

Previous studies have shown that, compared with ASCs, ASC microtissues can secrete more VEGF and nerve growth factor (NGF), which can affect adjacent damaged cells in a paracrine manner, and this may be more conducive to tissue repair [53,54]. Li et al. [55] used hanging drop methods to fabricate ASCs into microtissues to explore the efficacy and underlying mechanisms of three-dimensional ASC cultures for treating SUI. The microtissues were injected into periurethral tissue, and examinations were performed 28 days after transplantation. The leak point pressure (LPP) and bladder capacity of the microtissue treatment group were significantly higher than those of the ASC and control groups. The structure of the external sphincter was more complete, the content of total smooth muscle in the urethra was significantly increased, and the regeneration of angiogenesis and nerve regeneration was significantly accelerated. These results agree with the upregulated VEGF and NGF expression identified by immunofluorescence. Overall, transurethral injection of microtissues was more effective than ASCs in treating SUI. The experiment successfully tracked the distribution of injected ASCs using CM-Di labeling, but it was still difficult to observe the details of stem cell secretion and differentiation. The results also mentioned that there was no evidence of ASC differentiation into striated muscle cells. Thus, more durable and stable stem cell markers should be sought to establish a dynamic monitoring system to track the differentiation and secretion of ASCs in vivo.

Bulking agents based on synthetic and natural materials are effective in treating SUI; however, there is also the potential risk of immune reactions. Therefore, extracting ECM from autologous ASC sheets may help overcome this problem, whereby ASC sheets are incubated in 1% Triton X-100, 10 mM Tris, 1 μg/mL aprotinin, and 0.02% EDTA on an orbital shaker platform rotating at 120 rpm for 24 h. Wang et al. [56] implanted ECM fragments of autologous ASC sheets (ASC ECM) into the urethra of SUI rats. The ASC ECM fragments provided a bulking effect and induced muscle regeneration. Notably, ASC ECM retained its architectural morphology, major ECM-related components, and many growth factors, all of which are critical in promoting regeneration. A limitation of this experiment may be the lack of a comparable result of a bulking agent; although ASC ECM injection is promising for the regeneration of functional tissues, it is currently insufficient due to the lack of adequate amounts of cytokines to induce muscular regeneration, which are still not enough to restore the function of the urethral sphincter in the long term. Cell viability and retention rate at the graft site are critical for regenerative repair of the urinary sphincter. Wang et al. [57] subsequently attempted to control the aggregation of ASCs labeled with superparamagnetic iron oxide in the affected area by loading a magnetic field in vitro. Their results suggest that magnetic targeting can significantly improve the retention rate of stem cells aggregated in the affected area, and histological and urodynamic tests indicated that the structure and function of the sphincter muscle are significantly restored. This demonstrates that therapeutic effects may be improved by increasing the retention rate of transplanted stem cells in the affected area. The main issue with this study is that the optimal magnetic force and duration of magnetic targeting of cells were not elucidated in detail, and differences in treatment effects after changing the magnetic force and duration were not recorded. Furthermore, the 2-month evaluation period may not be long enough to observe the ultimate fate of injected ASCs. Inspired by studies that demonstrated that growth factors can provide optimal conditions for stem cell transplantation, Zhao et al. [58] combined ASCs and nerve growth factor encapsulated in poly lactic-co-glycolic acid microspheres into urethra and achieved adequate therapeutic effects. ASCs exhibited better cell viability with the help of nerve growth factor, which significantly improved LPP and the resting urethral pressure profile. Furthermore, muscle and peripheral nerve regeneration was accelerated. Overall, periurethral co-injection of ASCs and controlled-release NGF may be a potential strategy for SUI treatment.

ASCs combined with different kinds of bio-organic materials for the repair of urinary tract injury has always been a research hotspot. Epidemiological findings suggest that the increased incidence of SUI after menopause is likely to be associated with the decreased secretion level of 17-beta estradiol (E2), and artificial supplementation of E2 might reverse this effect. Thus, Feng et al. [59] inoculated pre-myogenic ASCs on a poly(L-lactide)/poly(e-caprolactone) electrospinning nano-scaffold. E2 was also incorporated into the system. The scaffold had good biocompatibility, E2 significantly accelerated stem cell proliferation, and α-SMA, calponin, and myosin heavy chain expression was significantly increased in the early, middle, and late stages of differentiation, respectively. Overall, these results suggest that the scaffold is a good material for tissue engineering and that E2 can promote the differentiation and proliferation of ASCs into smooth muscle cells. Experiments may be more complete if the pathways and underlying mechanisms of E2-promoting myogenic differentiation can be clearly elucidated. E2 upregulates MHC-I expression of muscle-derived stem cells and E2 receptor coactivators, and is likely to be one of the reasons for enhancing its transcriptional activity in vascular smooth muscle cells. Wang et al. [60] seeded ASCs on polyglycolic acid fibers as a suburethral sling for the treatment of SUI. ASCs formed tissue-engineered slings after 4 weeks in myoblast induction culture and maintained the myoblast phenotype within the sling, which contributed to the restoration of sphincter structure and function. The main limitation of the experiment is that no attempt was made to vary the mechanical loading to assess the effect because a constant strain without relief may not be the best way to exert mechanical loading stress. When periodic mechanical loading is applied in a bioreactor system, mechanical strength and tissue quality can be significantly improved. Therefore, dynamic strain may be a more appropriate method to attempt in the future.

Some research suggests that a large part of the therapeutic effect of stem cells is derived from their paracrine function rather than differentiation [61]. Nevertheless, the use of stem cells can lead to thrombosis, tumorigenicity, and immune rejection [62,63,64]. Therefore, stem cell-derived exosomes, which are important paracrine effectors in cell communication, may represent a safer cell-free therapy. For the treatment of SUI, Ni et al. [65] prepared ASC–exosomes using conditioned medium and ultracentrifugation and injected them into periurethral tissue. Cell Counting Kit-8 results indicated that ASC–exosomes can accelerate the proliferation of Schwann cells and muscle cells in a dose-dependent manner. Both bladder volume and LPP were significantly increased. Proteomics revealed that the signaling proteins carried by exosomes were closely related to the Wnt, JAK/STAT, and PI3K/AKT pathways. ASC–exosomes may play a therapeutic role by affecting the regeneration of muscles and nerves through these mechanisms. 

To elucidate this mechanism, Liu et al. [66] also made related attempts and found that ASC–exosomes could treat SUI. In vaginal fibroblasts from women with SUI, ASC–exosomes significantly upregulated TIMP-1, TIMP-3, and COL1A1 and downregulated MMP-1 and MMP-2 expression levels in fibroblasts in vitro, thereby increasing collagen synthesis and decreasing collagen degradation. Therefore, this study further suggests that ASC-derived exosomes might be a good treatment strategy for SUI. Inspired by the knowledge that *miR-93* can regulate collagen loss by targeting stromelysin-1 in human nucleus pulposus cells, Wang et al. [10] found that *miR-93* carried by ASC–exosomes can downregulate F3 (coagulation factor III) expression and simultaneously upregulate PAX7 and MyoD expression in fibroblasts. This in turn regulates ECM remodeling of SUI fibroblasts and activates satellite cells, achieving therapeutic effects. Table 3 is a summary of ASC-based therapy for stress urinary incontinence.

### 2.4. ASC-Based Therapy for Erectile Dysfunction

Erectile dysfunction (ED) often arises from multifaceted and complex mechanisms that may involve multiple causes, including blood vessels, neuropathy, and hormonal disruption [67]. ED has a high incidence and can seriously affect patient quality of life. Intracavernosal injections, penile implants, and vacuum erection devices can only control the state of the disease and cannot truly restore function [68]. ASCs can secrete neurotrophic factors, angiogenic factors, and reduce fibrosis, all of which have potential in treating ED [69].

Siregar et al. [70] suggested that the ability of ASCs to reduce normal tissue fibrosis may also be applicable to ED caused by collagen deposition in the corpora cavernosa. In the experiment, the injury model was constructed by 12 h of penile clamping and then treated with ASC injection. ELISA assay after 4 weeks revealed reduced TGF-β1 and type I collagen expression levels, and fibrotic changes were prevented. One of the limitations of this experiment is that it did not assess α-SMA and collagen type III, two indicators that may be meaningful in further research on fibrous tissue composition and inhibition of ASCs. Rather than using ELISA to detect tissue gene expression, PCR and Western blotting may be better options. In addition, adding an assessment of pathological and functional changes in corpora cavernosa may make the results more convincing. Castiglione et al. [71] further clarified the potential mechanism by which ASCs are capable of delaying fibrosis progression. The expression levels of 32 genes that regulate wound recovery, including *CXCL13*, *CXCR4*, *PLAT*, *SERPINH1*, and TGF-β1, were significantly changed after stem cell treatment. This may partly explain why ASCs can treat fibrosis-induced ED.

The retention rate of ASCs in the affected area is an important factor that influences the efficacy of stem cell therapy [72]. Higher retention rates may not only improve treatment outcomes but also reduce the likelihood of complications arising from injected stem cells escaping to other parts of the body. Zheng et al. [73] compared the effect of intracavernous injection of ASCs and ASC clusters in the treatment of neurogenic ED. Compared with direct ASC injection, the retention rate of ASC clusters in the corpus cavernosum was significantly higher, and the expression level of nerve and smooth muscle biomarkers was also upregulated. Additionally, the erectile function test suggests that intracavernous pressure/mean arterial pressure (ICP/MAP) and maximal ICP were more optimal in the ASC cluster treatment group. Furthermore, a caudal vein cluster injection group was added to explore whether the injection method influences treatment effects. Histopathological results indicated that the retention rate of intracavernous injection is much higher. Clusters injected through the caudal vein could not reach the corpus cavernosum, and most of them were found in lung tissue. The use of ASC self-aggregation is likely to be another direction for the treatment of ED, and the underlying mechanism may be to increase the retention rate of stem cells in the corpus cavernosum. This study also had some limitations. The study did not observe the dynamics of ASCs in rats after ASC transplantation to clarify their distribution and retention. Furthermore, self-aggregated cell clusters from ASC suspensions were used directly, and the number of cells they contained was not assessed, which also failed to generate the optimal therapeutic dose for ASC injection. Notably, Xu et al. [74] conducted similar experiments and obtained the same results.

Lipopolysaccharides can be recognized by Toll-like receptors, and Toll-like receptors can be expressed on the surface of ASCs [75,76]. Thus, Zhang et al. [77] pre-treated allogenic ASCs with lipopolysaccharides for the treatment of ED caused by cavernous nerve injury. Low doses of lipopolysaccharides can effectively improve the survival rate of ASCs and promote cell migration in vitro and inhibit caspase-3 activation induced by hydrogen peroxide, as well as downregulating TGF-β1 expression levels to delay smooth muscle fibrosis in vivo. Allogenic ASCs pretreated with lipopolysaccharides significantly increase smooth muscle content and alleviate penile fibrosis.

ASCs can be induced to differentiate into neural cells both in vivo and in vitro, and direct injection of ASCs into the cavernosa can promote the recovery of erectile function after nerve injury [78,79,80,81]. Thus, Ying et al. [82] cultured ASCs in vitro and undertook pre-neuralization treatment to induce ASC neural-like cells, which were used for ED caused by nerve damage. After neural-like cell transplantation, erectile function was better restored, and the maximal ICP, ICP/MAP, number of neuronal nitric oxide synthase-positive fibers, myelinated axons, and smooth muscle/collagen ratio were significantly increased. These findings suggest that ASC neural-like cells can effectively restore damage and improve erectile function. However, it is still unknown whether ASC neural-like cells can really be induced into nerve cells in vivo. ASC neural-like cells can promote the regeneration of nerve cells by accelerating the derivation of axons. Further research on growth factors released by the stem cells may be able to compensate for the lack of mechanism description to some extent. Zheng et al. [83] conducted a more detailed study and partially explained the mechanism. ASCs were pretreated with icariside II for differentiation into Schwann cells and then were used to treat ED caused by nerve injury. The erectile function of the rats was significantly restored. The reason may be that icariside II can inhibit *miR-33* overexpression, enabling the full function of glial cell-derived neurotrophic factor, promoting the differentiation of ASCs into Schwann cells and enhancing erectile function. Ge et al. [84] conducted similar experiments and found another possible pathway. Icariside II can also upregulate the expression levels of signal transducer and activator of transcription-3, which can decrease *miR-let-7i* expression, thereby releasing its mediated inhibitory effect on the differentiation of ASCs into Schwann cells.

Compared with the research on pre-differentiating stem cells in vitro, studies of genetic modification of ASCs to improve their efficacy in the treatment of ED have increased more gradually in recent years. Corpus cavernosum smooth muscle can control the blood flow into the corpus cavernosum and plays an important role in the process of erection [85]. Furthermore, myocardin is an important factor that maintains the function of smooth muscle cells [86,87]. Zhang et al. [88] used myocardin- transduced ASCs to treat a rat model of ED caused by diabetes. The transduction of the myocardin gene, *MYOCD*, had no obvious effect on cell apoptosis but inhibited cell proliferation and promoted cell contraction. This transduction significantly upregulated calponin and α-SMA expression and increased both maximal ICP and ICP/MAP, thereby enhancing the therapeutic potential of ASCs. The use of adenoviral vectors in the experiment made the longest observation time to be 3 weeks. Of course, it is understandable that adenoviral vectors are currently selected to satisfy both the high transfection efficiency and cotransfection with EdU (5-ethynyl-2′-deoxyuridine), but it may be a good direction to use lentiviral vectors to monitor long-term effects.

Glial cell line-derived neurotrophic factor (GDNF) can trigger stem cell migration and improve neuronal survival [89], and VEGF can promote vascular regeneration [90]. Therefore, Yang et al. [91] evaluated the effect of ASCs overexpressing GDNF and VEGF in the treatment of ED caused by nerve injury. At 2 weeks after transplantation, the erectile function of the rats was almost completely restored. Compared with ASCs with single gene overexpression, ASCs with both GDNF and VEGF overexpression significantly improved cavernous nerve repair, vascular endothelium regeneration, and penile fibrosis. Overexpressed GDNF and VEGF can exert a synergistic effect and improve the therapeutic effect of ASCs. Yang et al. [92] further found that ASCs can significantly increase IGF-1, bFGF, and VEGF expression in penile tissues as well as the number of cavernous smooth muscle cells and the ICP/MAP of the treated aged rats. These results suggest that ASCs can upregulate IGF-1, bFGF, and VEGF expression to treat aging-induced ED.

Genetic modification is also a potential approach to better exert the therapeutic effect of ASCs. Zhou et al. [93] found that *miR-423-5p* could inhibit endothelial nitric oxide synthase (NOS) and *VEGFA* gene expression. Thus, they designed and successfully optimized the effects of ASCs in the treatment of diabetes mellitus ED by knocking out *miR-423-5p*. After transplantation of *miR-423-5p*-knockout ASCs, ICP/MAP was significantly increased, and erectile function was improved. The expression of endothelial NOS and VEGF was significantly upregulated after treatment, potentially explaining why *miR-423-5p* knockout can improve the therapeutic effects of ASCs. The experiment did not assess the fibrosis factors that lead to ED. In addition, ASCs and human umbilical vein endothelial cells were co-cultured in the experiment. The underlying mechanism by which the two interact may be described in future studies. Inspired by the important role of brain-derived neurotrophic factor (BDNF) in promoting nerve regeneration and treating ED, Yang et al. [94] achieved better therapeutic effects by transducing ASCs with recombinant adenoviral vectors expressing BDNF. After ASC–BDNF treatment, erectile function was significantly recovered and the content of neuronal NOS and ratio of smooth muscle/collagen in the dorsal nerve were significantly increased. These results indicate that ASC–BDNF can effectively improve ED caused by nerve injury. However, the experiment still had some limitations. For example, the safety of the lentivirus that introduced BDNF into the ASCs is not yet clear. Furthermore, the whole experiment lasted only 4 weeks, and it is difficult to evaluate the long-term efficacy of ASC–BDNF on neurogenic ED.

Several studies have shown that the transplantation of endothelial progenitor cells can be used to treat hindlimb [95] and coronary ischemia [96]. However, endothelial cells are severely damaged in diabetes, and pro-angiogenic factors required for their proliferation cannot be synthesized by themselves [97,98], but can be provided by stem cell paracrine factors. Guo et al. [99] discovered that genetically modified endothelial progenitor cells can be used to treat ED. Thus, Yang et al. [100] progressed a step further and treated diabetes-induced ED by combining ASCs with endothelial progenitor cells. Stromal-derived factor-1 and VEGF secreted by ASCs increased significantly, which also resulted in endothelial progenitor cells expressing higher levels of 5-ethynyl-2′-deoxyuridine in the cavernous endothelial layer, promoting repair. The combined treatment significantly increased ICP and ICP/MAP, which is likely to be closely related to the correction of the eNOS/cGMP/NO signaling pathway and the high expression of the corresponding CD31. The main limitation of the experiment is that the CD31 and EdU double-positive cells used in combination with ASCs are not necessarily endothelial progenitor cells, but may also be mature endothelial cells, and a single marker cannot distinguish them. The overall duration of the experiment was relatively short, and the long-term efficacy of the combination therapy may still need to be assessed because hyperglycemia is likely to impair the function of ASCs and endothelial progenitor cells over time, thereby making the treatment less effective.

Some researchers claim that the paracrine function, rather than the proliferation, of stem cells plays a large role in tissue repair, and that cell-free therapy with paracrine signaling may be a safer and more effective treatment strategy. Wang et al. [101] used ASC–exosomes for the treatment of diabetes-induced ED and achieved good results. ICP/MAP was increased (representing an improvement in erectile function), nerve and blood vessel regeneration was accelerated, and atrial natriuretic peptide, brain natriuretic peptide, and neuronal NOS expression were inhibited. This is probably related to the effect of the *CORIN* gene, because after its silencing, the effect of ASC–exosomes in the treatment of ED was greatly reduced. Liang et al. [72] prepared polydopamine nanoparticles incorporated in poly(ethylene glycol)-poly(ε-caprolactone-co-lactide) thermosensitive hydrogels for intratunical administration of ASC–exosomes. The hydrogels can cause exosomes to exhibit sustained release behavior. Animal experiments indicated that exosomes released from the hydrogel can more effectively promote the regeneration of endothelial and nerve cells (by increasing the expression of eNOS and nNOS), increase cavity pressure, and thus restore erectile function. In addition, the polydopamine nanoparticles in the hydrogel have good photoacoustic performance, thereby enabling hydrogel-carrying exosomes to be accurately transported into the tunica albuginea by real-time photoacoustic imaging. Table 4 is a summary of ASC-based therapy for erectile dysfunction.

## 3. Current Challenges and Future Directions

To date, research using ASCs to repair and reconstruct the lower urinary tract has covered many aspects, including induced differentiation, gene modification, microenvironment regulation, paracrine pathways, exosomes, and cell sheets and their derivatives (Figure 2). The focus of research has gradually shifted from demonstrating the safety and efficacy of ASCs to revealing the underlying mechanisms of ASCs that can promote the recovery of tissue and function and improve the efficacy of ASC treatment. These results are encouraging; however, some fundamental questions remain to be clarified.

The optimal treatment dose of ASCs was not investigated in most experiments, and the relationship between the severity of injury and the number of transplantations remains to be further elucidated. Currently, no optimal method has been established for ASC delivery, which most likely influences whether stem cells can reach the affected area and their retention rate [57,102]. Fortunately, some studies have focused on this issue. It is now theorized that the therapeutic effect of stem cells on damaged tissues is largely derived from their signal transduction effects [103,104,105], and many studies are focused on enhancing the treatment effect by strengthening these pathways.

Cell-free therapy using exosomes may be a future direction. However, at present, such therapies still present many challenges: the exosome clearance rate is high, the half-life of exosomes is short, the activity of free exosomes is difficult to maintain, and regeneration often takes a long time [106]. Because of their inherent homing ability, ASC–exosomes are very promising as stable and effective carriers. Careful design of these carriers could enable loading of specific genetic material, lipids, and proteins, as well as selective transport to target tissues or organs [107,108].

Precision therapy is clearly a constant pursuit. Fibroblasts and myofibroblasts play different roles in different stages of tissue reconstruction, and they often show huge functional differences between different tissues [109]. Therefore, to enhance the therapeutic effects of stem cells, clarifying the regulation of fibroblasts and myofibroblasts by ASCs and ASC–exosomes during different healing stages is an urgent research direction. Stem cells also have the ability to undergo abnormal differentiation and malignant transformation. Thus, whether stem cell transplantation causes genomic or epigenetic changes and enduring immune responses [110,111,112] must be investigated to determine their long-term safety.

## Figures and Tables

**Figure 1 pharmaceutics-14-02229-f001:**
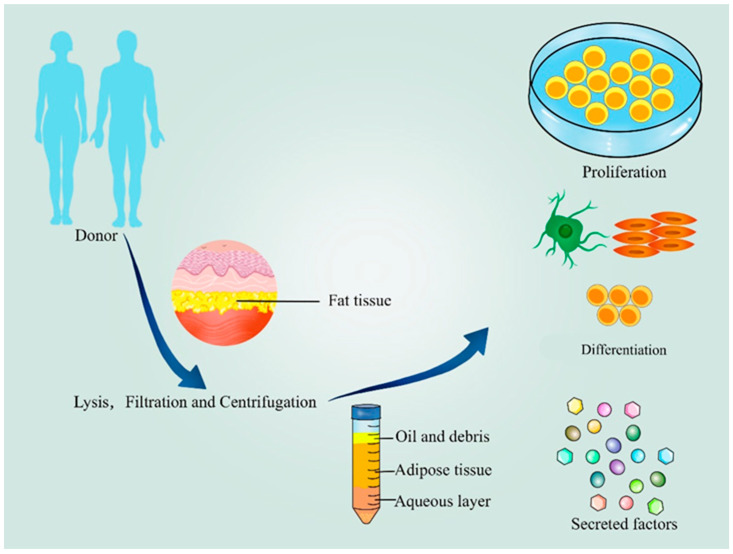
Extraction, proliferation, differentiation, and secretion of ASCs.

**Figure 2 pharmaceutics-14-02229-f002:**
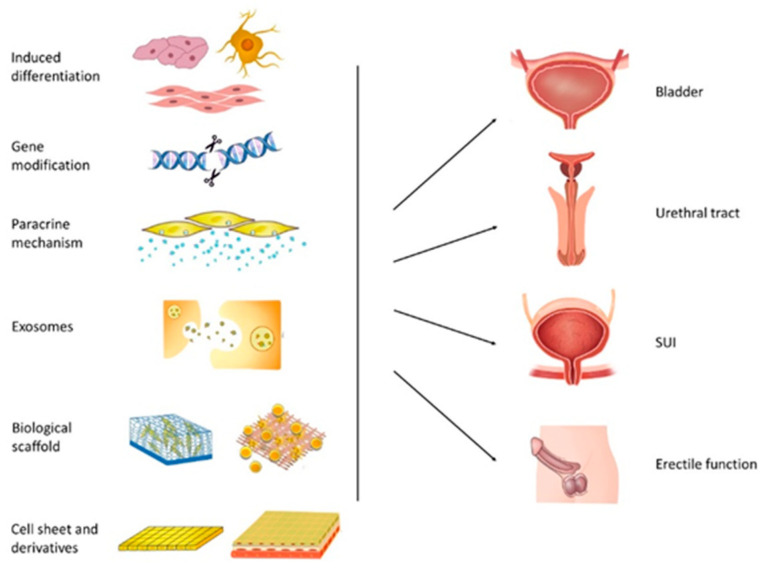
Approaches to enhance ASC therapeutic performance for lower urinary tract dysfunction.

**Table 1 pharmaceutics-14-02229-t001:** Overview of ASC-based therapy for bladder reconstruction.

Authors and Year	Treatment Strategy	Biological Effect
Zhe et al., 2016	ASC-seeded BAMGs	Increases in collagen bundles, myofibroblasts, and nerve cell regeneration.
Pokrywczynska et al., 2018	ASC-seeded BAMGs	ASCs differentiated into smooth muscle.
Hou et al., 2016	ASC-seeded BAMGs	Urothelium, bladder smooth muscle, and capillary vessel regeneration.
Moreno-Manzano et al., 2020	ASC-seeded BAMGs	ASC-seeded BAMGs also accelerated angiogenesis and neuronal regeneration.
Xiao et al., 2017	ASC-seeded BAMGs and silk fibroin	Expression levels of SDF-1α, VEGF, and their receptors were upregulated, and ERK 1/2 phosphorylation was increased.
Xiao et al., 2021	ASC-seeded BAMGs and silk fibroin	ASCs exerted pro-angiogenic effects through the SDF-1α/CXCR4 pathway and its downstream ERK1/2 phosphorylation.
Xiao et al., 2017	ASCs-seeded BAMGs and alginate di-aldehyde-gelatin hydrogel-silk mesh	Smooth muscle regeneration, angiogenesis, and innervation of the bladder were significantly accelerated.
Wang et al., 2019	ASC sheet and porous polyglycolic acid scaffold with ultra-small superparamagnetic iron oxide nanoparticles	ASC sheets promoted urothelial cell, smooth muscle cell, and neurovascular regeneration.
Yang et al., 2021	ASC–exosomes and BAMGs	Angiogenesis was more pronounced with ASC–exosome-stimulated BAMGs.
Xiao et al., 2021	ASC–exosomes and BAMGs alginate dialdehyde-gelatin hydrogel	By upregulating *miRNA-12* expression levels, inhibiting RGS16 to activate the CXCR4/SDF-1α pathway ultimately increased VEGF secretion.

**Table 2 pharmaceutics-14-02229-t002:** Overview of ASC-based therapy for urethral injuries.

Author and Year	Treatment Strategy	Biological Effect
Castiglione et al., 2016	ASCs	Fibrosis-associated genes were significantly reduced.
Feng et al., 2018	ASCs and *miR-21*	*MiR-21*-modified ASCs increased the expression levels of the angiogenic factors hypoxia inducible factor-1, VEGF, bFGF, stem cell factor, and stromal derived factor-1a.
Rashidbenam et al., 2021	ASC sheet and urothelial and smooth muscle cells	ASC sheet seeded with urothelial and smooth muscle cells was used as a scaffold.
Zhou et al., 2017	ASC sheet and fibroblast and myoblast differentiated cell sheets	ASC sheet for construction of a three-layer bionic urethra.
Sa et al., 2018	Pre-epithelialized ASC-seeded BAMGs	Post-transcriptional inhibition of TIMP-1 by *miR-365* in ASCs could further inhibit fibrosis.
Li et al., 2014	Pre-epithelialized ASC-seeded BAMGs	Pre-epithelialized ASC-seeded BAMGs better restored a continuous epithelial layer.
Fu et al., 2014	ASC sheet and oral mucosal epithelium and poly-glycolic acid	Myogenic differentiated ASCs induced by mechanical extension were combined with oral mucosal epithelium to form an artificial urethral substitute with a complete double-layer structure.
Tian et al., 2018	ASCs and silk fibroin	Six to seven layers of urothelial cells were formed on the surface of the silk fibroin, and the numbers of smooth muscle cells and new blood vessels growing along the silk fibroin pores were significantly increased.
Wang et al., 2022	ASCs-exosomes and collagen/poly (L-lactide-cocaprolactone) nanoyarn scaffold	ASC–exosomes nanoyarn scaffold significantly promoted epithelialization and vascularization and accelerated the transition of damaged tissues from an inflammatory state to a regenerative state.

**Table 3 pharmaceutics-14-02229-t003:** Overview of ASC-based therapy for stress urinary incontinence.

Author and Year	Treatment Strategy	Biological Effect
Tehrani et al., 2021	ASCs and muscle-derived stem cells	The striated muscle and external sphincter around the urethra were significantly thickened, and the urethral pressure curve was significantly higher.
Li et al., 2016	ASC microtissues	The structure of the external sphincter was more complete, the content of total smooth muscle was increased, and the regeneration of angiogenesis and nerve regeneration was accelerated.
Wang et al., 2020	ECM from ASC sheet	ASC sheet ECM fragments provided a bulking effect and induced muscle regeneration.
Wang et al., 2020	ASCs labeled with superparamagnetic iron oxide	Magnetic targeting can improve the retention rate of stem cells aggregated in the affected area.
Zhao et al., 2011	ASCs and nerve growth factor encapsulated in E2 incorporated poly lac-tic-co-glycolic acid microspheres	Periurethral co-injection of ASCs and controlled-release NGF accelerated muscle and peripheral nerve regeneration.
Feng et al., 2016	Pre-myogenic ASCs and poly(L-lactide)/poly(e-caprolactone) nano-scaffold	E2 significantly accelerated stem cell proliferation. Furthermore, α-SMA, calponin, and myosin heavy chain expression was increased in the early, middle, and late stages of differentiation, respectively.
Wang et al., 2017	ASCs and polyglycolic acid fibers	ASCs formed tissue-engineered slings that exhibited improvements in biomechanical properties, and the tissue and collagen structures matured.
Ni et al., 2018	ASCs–exosomes	The signaling proteins carried by ASCs exosomes were closely related to the Wnt, JAK/STAT, and PI3K/AKT pathways.
Liu et al., 2018	ASC–exosomes	ASC–exosomes could upregulate TIMP-1, TIMP-3, and COL1A1 and downregulate MMP-1 and MMP-2 expression levels in fibroblasts, thereby increasing collagen synthesis and decreasing collagen degradation.
Wang et al., 2021	ASC–exosomes	ASC–exosomes could downregulate F3 expression and upregulate PAX7 and MyoD expression in fibroblasts.

**Table 4 pharmaceutics-14-02229-t004:** Overview of ASC-based therapy for erectile dysfunction.

Author and Year	Treatment Strategy	Biological Effect
Siregar et al., 2020	ASCs	ASC injection reduced TGF-β1 and type I collagen expression levels.
Castiglione et al., 2019	ASCs	The expression levels of genes regulating wound recovery, including CXCL13, CXCR4, PLAT, SERPINH1, and TGF-β1, were significantly changed after stem cell treatment.
Zheng et al., 2021	ASC clusters	The retention rate of ASC clusters in the corpus cavernosum was significantly higher than ASC injection.
Zhang et al., 2022	Lipopolysaccharide pre-treated ASCs	Low doses of lipopolysaccharides could improve the survival rate of ASCs, inhibit caspase-3 activation induced by hydrogen peroxide, promote cell migration, and downregulate TGF-β1 expression levels to delay smooth muscle fibrosis.
Ying et al., 2019	Pre-neuralization ASCs	ASC neural-like cell transplantation improved erectile function through increasing the number of neuronal nitric oxide synthase-positive fibers, myelinated axons, and smooth muscle/collagen ratio.
Zheng et al., 2020	ASCs and *miR-33* inhibitors (Icariside II)	Icariside II promoted the differentiation of ASCs to Schwann cells via regulating *miR-33*.
Ge et al., 2019	ASCs and *miR-33* inhibitor (Icariside II)	Icariside II can upregulate the expression levels of signal transducer and activator of transcription-3.
Zhang et al., 2019	Myocardin-transfected ASCs	Transfection of *MYOCD* had no obvious effect on cell apoptosis but inhibited cell proliferation and promoted cell contraction.
Yang et al., 2018	GDNF- and VEGF-overexpressing ASCs	ASCs with both GDNF and VEGF overexpression better improved cavernous nerve repair and vascular endothelium regeneration.
Yang et al., 2020	ASCs	ASCs could increase IGF-1, bFGF, and VEGF expression in penile tissues, the number of cavernous smooth muscle cells, and ICP/MAP of the treated aged rats.
Zhou et al., 2021	*miR-423-5p*-knock-out ASCs	*miR-423-5p*-knockout ASCs relieved its inhibition on endothelial *NOS1* and *VEGFA* gene expression with therapeutic benefits.
Yang et al., 2020	BDNF-overexpressing ASCs	ASCs infected with lenti-rBDNF increased the number of nNOS-positive nerve fibers and smooth muscle in the penile tissue, thereby improving ED caused by nerve injury.
Yang et al., 2020	ASCs and endothelial progenitor cells	Stromal-derived factor-1 and VEGF secreted by ASCs increased significantly, which also resulted in endothelial progenitor cells expressing higher levels of 5-ethynyl-2′-deoxyuridine. This promoted repair.
Wang et al., 2020	ASCs–exosomes	After ASC exosome injection, ICP/MAP was increased, nerve and blood vessel regeneration were accelerated, and atrial natriuretic peptide, brain natriuretic peptide, and neuronal nitric oxide synthase expression was inhibited.
Liang et al., 2022	ASCs–exosomes	Polydopamine nanoparticles incorporated in hydrogels could cause ASC–exosomes to exhibit sustained release behavior. Furthermore, they could more effectively promote the regeneration of endothelial and nerve cells by increasing the expression of eNOS and nNOS.

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
