# Peer review of "Therapies Based on Adipose-Derived Stem Cells for Lower Urinary Tract Dysfunction: A Narrative Review"

_pharmaceutics, 2022, doi:10.3390/pharmaceutics14102229_

Round 1
Reviewer 1 Report
The manuscript ‘Therapies based on adipose-derived stem cells for lower urinary tract dysfunction: a narrative review’ introduces that ADSCs is considered a promising therapeutic strategy for lower urinary tract repair and reconstruction. Biotechnology is within the intended scope of this journal. The introduction however lacks the why? The argument to write this review is missing. All the subtopics are already subjected to more or less recent reviews and it e.g. does not become clear what this review adds. (and the citation of ref 1 includes given names, not family names) I can appreciate its content and it’s four sub-topics although I am not certain that erectile dysfunction is generally considered ‘lower urinary tract dysfunction” (and this erectile dysfunction is not mentioned in the abstract and, the abstract also says ‘treating LUT diseases’. This should be dysfunctions, since no diseases are included. This is a narrative review, so I can accept that the articles are expert selected and not ‘systematically selected’ however it is not explained why you found /selected 55 studies for ‘qualitative analysis’ while the review contains almost double the number of references. I appreciate that studies-selection is better explained with replicable criteria. In general I ask to consider to add on to earlier reviews and then conclude what progress is made in the specific field of research. Yes, these techniques are promising, but are they just as promising as they were 20 years ago? ADSC + ‘promising’ hits 2440 results today on my PubMed search and 3,737 hits on scopus and thus almost 1/5 of the studies has use this word in the abstract…. This is of course a crude search with high chance of bias, but is it publishable nowadays to just conclude that the results ‘are encouraging’? I very much would have appreciated to read which fundamental questions remain to be clarified. This (last) sentence (of the conclusion) should have been in the introduction: If each of the paragraphs had given an answer to these (now un –asked) questions on the basis of a transparent selection of manuscripts, I would have wholeheartedly recommended publication.
Reviewer 2 Report
This review provides a summary of major studies in the important field of regenerative medicine to address urological defects and trauma, with a focus on the use of adipose-derived stromal/stem cells.
It is informative and interesting to read. However, there has been truly a LOT of reviews recently published for this specific topic, so in order to bring originality, several changes would benefit the manuscript:
1) The authors are strongly encouraged to use the term ASCs throughout the manuscript and not ADSCs, since ASC was the term selected by the discoverer of ASCs and endorsed by the IFATS society. (Daher et al. 2008, STEMCELLS 2008;26:2664–2665 www.StemCells.com)
2) It is also suggested to use the term “post-natal” instead of adult stem cells, since autologous cells can be derived from children, which often suffer from urological defects at birth.
3) It is also suggested to use the term “adipose tissue” more often in replacement of “fat” which refers more to lipids.
4) The review shoud include Tables to provide more structure and importantly provide more details on the studies cited, which are often presented in a very concise fashion preventing to fully analyze and compare the literature adequately.
5) Figure 2 provides a flow diagram, but the time period looked at (?5 years, 10 years? More?) is not mentioned and should be provided. Also the keywords used for this search are not provided. If the authors used ADSC instead of ASC, they might have missed significant papers.
6) Page 2 of 17, lines 47 onwards: There are a lot of concepts in this paragraph. It is suggested to address the concept of exosomes if a different sentence/section than cells, to convey a simpler message. Also, exosomes will need to be introduced more properly (definition, how to obtain, content), which is currently missing in the text.
7) Please correct typo in Figure 1 (differentiation)
8) Page 3 of 17 line 72, please add a reference concerning the gastrointestinal segment.
9) Line 75: “their efficacy in bladder regeneration has also been proven”… ?in preclinical studies? as a general comment: very often, it lacks context, the context of the research should be specified more: preclinical data, or clinical studies?. Also would be useful to mention how many fold increases achieved when a treatment is deemed to have worked by enhancing this or this process?
10) Line 89: please specify which type of imaging modality was used.
11) Line 235: please provide more context to the expression” preepithalialized ASC” which can be confusing and not widely known how they cells were obtained.
12) Line 122: In tissue extracts? or tissue sections, how?
As mentioned previously, more details could be provided to the reader while describing each study. Thus the usefulness of tables to provide more parameters, methods and results without lengthening the text itself as much.
13) Line 138: please explain, or change since detaching ASCs from culture flasks has not been reported to alter their properties or functionality when done properly !
14) Line 152: more explanations on how these tissues were generated are needed.
15) Line 191: closer to normal levels
16) Line 313: immune reaction
17) The SUI section reads well and better describe the studies. However line 362 onwards, context should be precised: in vitro only? The conclusions on therapeutic effects should be toned down a little if in vitro studies only were carried out.
18) Lines 406-413, please provide more precisions, which results/conclusions are in vitro versus in vivo, etc.
19) Lines 432-434: ? has increased more gradually.
20) Line 463: transducing with recombinant adenoviral vectors
21) Of course, it would have been interesting to know more clearly after finishing to read this manuscript, what modalities are currently in clinical trials in humans. This was not addressed.
Reviewer 3 Report
The authors submitted a manuscript summarizing the effect of therapies based on adipose-derived stem cells (ADSCs) on the treatment of lower urinary tract dysfunction.
Lower urinary tract diseases are emotionally and financially burdensome to the individual and society. Current treatments are ineffective or symptomatic. Conversely, stem cells are regenerative and may offer long-term solutions. However, results have been conflicting because of the variability in cell numbers, biomaterials types, and graft surface differences. ADSCs can be easily and abundantly obtained from "discarded" adipose tissue. ADSCs have self-renewal and multi-differentiation potentials, which can differentiate into smooth muscle cells, urinary epithelial-like cells, endothelial cells, neuron-like cells, etc. and secrete a variety of growth factors.
It is very important to note that the review should not be simply a description of what others have published in the form of a set of summaries, but should take the form of a critical discussion, showing insight and an awareness of differing arguments, theories and approaches. However, in this manuscript, the authors simply present and list the results that have been published in the past with offering few insights of their own. ADSCs research is gaining wide attention and the research on ADSCs in lower urinary tract reconstruction using tissue engineering has been a highlight in recent years. There have been already similar articles. Moreover, this manuscript focuses on ADSCs rather than bio-organic materials, and more than half of the manuscript content has nothing to do with the theme of the special issue (Bio-Organic Materials for Tissue Engineering and Regenerative Medicine). This can also be reflected in the title and keywords.
Some specific comments are listed below.
1. "Data acquisition" is not appropriate as the title of the entire second section.
2. It would be better to have more summary tables in the manuscript.
3. The English needs to be improved to a certain extent. There are some errors in grammar and format in the whole manuscript: inconsistencies; single and plural expressions; the use of prepositions and definite/indefinite articles.
Reviewer 4 Report
Comments on pharmaceutics-1846619
In this review, Liu and coworkers summarize the most recent studies using cell therapy and tissue engineering, with a focus on adipose stromal cells, for lower urinary tract dysfunctions. In their introduction, the authors argue that the use of adipose-derived stem cells (ADSCs) poses promising source cells for genitourinary regenerative therapies. Next, they describe the process of literature screening and choosing, under the title "Data acquisition". Then, the sub-titles are organized by the clinical challenge the therapy is aimed to treat: ADSC-based therapy for bladder reconstruction, …for urethral injuries, …for stress urinary incontinence (SUI) and last, …for erectile dysfunction. The last chapter, titled "Current challenges and future directions", covers some of the unanswered questions in the field and mentions several general weak points in the studies in an unspecific manner.
The review is timely and well-written, however, four points should be addressed:
· The introduction (which should be titled "rational for focusing on ADSC" or something like that) does not give an overview of the different sources of cells available for cell therapy of lower urinary tract dysfunctions. Although the decision to focus on one type of stem cell is justified, several sentences describing other options should be added.
· The chapter "data acquisition" indicates that the review is a formal meta-analysis of clinical trials. This is not the case, and the standard form of a scientific review does not require this type of literature screening description, especially due to the fact that no scientific standards were made, (every research article in English was included) so the general impression here is a bit ridiculous.
· Organizing the review by the medical problem point again to the medical/clinical orientation of the authors. This is fine, but the review should be at least readable and somewhat comprehensible to scientists who are not M.Ds, so it is advisable to add a short description of the clinical problem and solutions available, before diving into the relevant cell-therapy experiment.
· The content, e.g. description of the studies, refrain from pointing out their limitations, flaws and shortcomings. This is the major limitation of this review…
· No attempt to organize the data in any way useful to the reader is made, therefore the manuscript reads as a long list of study descriptions, without summary sentences that put them into a broader context, and without any connections and comparisons between the different experimental strategies done by the authors. In short, the manuscripts lack a scientific perspective and authority that will allow the reader to gather meaningful conclusions.
Minor points:
· Figure 3 adds nothing to the paper.
· A big portion of the cited studies were done in Shanghai, some are from the same hospital, institute or university that some of the authors are affiliated with. This is not an ethical problem but it raises the question of whether the authors are aware enough of studies performed in other countries. In support of that, in a parallel review recently published1 I could not detect a similar focus on studies from shanghai.
Overall, I believe this review should be published after an extensive effort into re-writing, drawing figures and tables and a more detailed introduction is made, toward better readability and clarity for a broader audience.
1Caneparo C, Sorroza-Martinez L, Chabaud S, Fradette J, Bolduc S. Considerations for the clinical use of stem cells in genitourinary regenerative medicine. World J Stem Cells. 2021 Oct 26;13(10):1480-1512. doi: 10.4252/wjsc.v13.i10.1480.
Round 2
Reviewer 1 Report
I consider the manuscript very much improved, my compliments to the authors. I have only one very minor comment. The word 'often' is used in the added sentence, I recommend to replace this with 'may'. (Or to refer to literature that precisely reports (complication) frequencies of all treatment possibilities, but that would be too much regarding the scope of the manuscript.) I now consider this a typing error.
Reviewer 2 Report
In general, the authors have provided additional information that ameliorated the manuscript and answered my initial requests. In addition, the new Tables help to synthesize the conclusions.
Please perform these minor corrections on this revised version:
1-The definition of exosomes in not complete enough still: please indicate size, surface markers, how to they are produces by cells (exosomes are a subfamily of extracellular vesicles…)
2-Please add references to new paragaph at lines 43-49
3-New Lines 65-66, please remove as there are quite many reviews on the topic.
4-Many misspelling errors can be found in the newly added text, please revise carefully.
5-Please correct the use of ADSCs for ASCs in the tables (column Biological effects) and newly added text throughout the manuscript.
6-Line 300: please remove “ bionic”
7- Lines 574 transduced instead on transfected…, lines 579 onwards, adenovirus = adenoviral vectors, lentiviral vectors…
8- Line 581, please define EdU.
Reviewer 3 Report
The authors submitted a revised manuscript and made some corresponding changes highlighted based on the reviewers' comments.
Nevertheless, there are still some issues in the manuscript.
1. The authors mentioned “finally decided to delete ‘Data acquisition’”, so what exactly is the title of the second section now?
2. The tables should be in the form of a three-line table.
3. Although the authors mentioned “The paper has been revised by an editor whose native language is English, and grammar errors have been corrected.” in the response, the revised manuscript still contains some errors. For examples:
In Line 147, “Eperiments” should be changed into “Experiments”;
TGFβ1, TGFb1, TFG-β1;
…
Reviewer 4 Report
The authors have revised the manuscript extensively and it is now suitable for publication. the sentences added at the end of most paragraphs are of great value to the reader
Round 3
Reviewer 3 Report
The authors submitted a revised manuscript and made some corresponding changes highlighted based on the reviewer's comments. Some changes are not marked. The tables are still not in the format of a three-line table.
